# Geniposide ameliorates bleomycin-induced pulmonary fibrosis in mice by inhibiting TGF-β/Smad and p38MAPK signaling pathways

**Jian-Bin Yin[1,2☯], Ying-Xia Wang[3☯], Su-Su Fan[2☯], Wen-Bin Shang[2], Yu-Shan Zhu[2], Xue-Rong Peng[2], Cheng Zou[2]\*, Xuan Zhang[2]\***

**1** The People's Hospital of ChuXiong Yi Autonomous Prefecture, ChuXiong, China, **2** School of Pharmaceutical Sciences & Yunnan Key Laboratory of Pharmacology for Natural Products, College of Modern Biomedical Industry, Kunming Medical University, Kunming, China, **3** Department of Pathology, the First Affiliated Hospital of Kunming Medical University, Kunming, China

☯ These authors contributed equally to this work.
\* zouchengkm@126.com (ZC); snoopykm@126.com (XZ)

**Data Availability Statement:** All relevant data are within the manuscript and its Supporting Information files.

## Abstract

Pulmonary fibrosis (PF) is an interstitial lung disease characterized by inflammation and fibrotic changes, with an unknown cause. In the early stages of PF, severe inflammation leads to the destruction of lung tissue, followed by upregulation of fibrotic factors like Transforming growth factor-β (TGF-β) and connective tissue growth factor (CTGF), which disrupt normal tissue repair. Geniposide, a natural iridoid glycoside primarily derived from the fruits of Gardenia jasminoides Ellis, possesses various pharmacological activities, including liver protection, choleretic effects, and anti-inflammatory properties. In this study, we investigated the effects of Geniposide on chronic inflammation and fibrosis induced by bleomycin (BLM) in mice with pulmonary fibrosis (PF). PF was induced by intratracheal instillation of bleomycin, and Geniposide(100/50/25mg•kg$^{-1}$) was orally administered to the mice once a day until euthanasia(14 day/28 day). The Raw264.7 cell inflammation induced by LPS was used to evaluate the effect of Geniposide on the activation of macrophage. Our results demonstrated that Geniposide reduced lung coefficients, decreased the content of Hydroxyproline, and improved pathological changes in lung tissue. It also reduced the number of inflammatory cells and levels of pro-inflammatory cytokines in bronchoalveolar lavage fluid (BALF) of bleomycin-induced PF mice. At the molecular level, Geniposide significantly down-regulated the expression of TGF-β1, Smad2/3, p38, and CTGF in lung tissues of PF mice induced by bleomycin. Molecular docking results revealed that Geniposide exhibited good binding activity with TGF-β1, Smad2, Smad3, and p38. In vitro study showed Geniposide directly inhibited the activation of macrophage induced by LPS. In conclusion, our findings suggest that Geniposide can ameliorate bleomycin-induced pulmonary fibrosis in mice by inhibiting the TGF-β/Smad and p38MAPK signaling pathways.

**Funding:** This research was funded by the National Natural Science Foundation of China (No. 82260727),Yunnan Provincial Science and Technology Department (No.202101AY070001-010), and the Innovation Team Construction Project of Kunming Medical University (CXTD202203).

**Competing interests:** The authors have declared that no competing interests exist.

## 1. Introduction

Pulmonary fibrosis (PF) is a progressive interstitial lung disease caused by a variety of causes. Due to inflammation and subsequent scar formation, the alveolar structure is disrupted, lung compliance is reduced, gas exchange is interrupted, and this eventually leads to respiratory failure and death [1, 2]. It is worth noting that the global incidence of PF seems to increase over time, with a conservative estimate of 3–9 per 100,000 people per year, and the median survival after diagnosis is only 2–4 years [3].

The clinical manifestations of PF patients are mainly dry cough, dyspnea on exertion, and overall progressive deterioration of the patient's quality of life (QOL). Over the past decade, awareness of the pathogenesis and management of PF has changed [4], and two drugs (pirfenidone and nintedanib) have been marketed and used for treating PF; unfortunately, the hepatotoxicity and gastrointestinal side effects of the two drugs are severe, which limits their clinical application. In addition, although treatments with both drugs slowed the decline in forced vital capacity (FVC), the quality of life (QOL) did not improve in patients with PF [5, 6]. Therefore, the treatment of PF is still a difficult problem in clinics, and the need for developing new drugs with high effects and low toxicity for PF is still urgent.

Bleomycin (BLM) is a glycopeptide antibiotic with potent anti-tumor activity and is mainly used for the treatment of various cancers and lymphomas. Because of the advantages of easy induction and reproducibility of PF model induced by bleomycin, it has been widely used in researching the pathogenic mechanism of PF-related diseases and identifying new therapeutic targets [7]. Even though the pathogenesis of PF is still unclear, the general view is that the development of PF is accompanied by inflammation. Studies have confirmed that the progression of PF patients is related to the degree of inflammation [8], which is mainly dominated by inflammatory cells and inflammatory factors, such as lymphocytes [9], intrapulmonary-derived macrophages [10] and tumor necrosis factor-α (TNF-α) [11]. TGF-β1 in the Transforming growth factor-β (TGF-β) family is a key cytokine involved in the formation of pulmonary fibrosis [12], which is referred to as the "Total switch" of cytokines. TGF-β1 plays an essential role in fibrotic processes, it can stimulate the fibroblasts and macrophages to synthesize proinflammatory and fibrogenic factors such as TNF-α, platelet-derived growth factor (PDGF) [13], interleukin-1β (IL-1β), and IL-13 [12]. TGF-β1 signaling pathways are mainly divided into small mother against decapentaplegic (Smad)-dependent and non-Smad-dependent pathways.

The TGF-β1-mediated Smad-dependent signaling pathway is not negligible during PF. It can transfer exogenous stimuli into the nucleus to regulate the transcription of downstream target genes. The possible mechanism is that TGF-β1 binds to its receptor, activates the receptor-regulated Smad2 and Smad3, and then forms a trimer with Samd4 to translocate into the nucleus [14]. Connective tissue growth factor (CTGF) is secreted by active fibroblasts and epithelial cells and is a downstream effector of the TGF-β1 signaling pathway, linking TGF-β1 to the production of extracellular matrix (ECM) in PF. It is worth noting that the TGF-β1/Smad pathway is also involved in the activation of CTGF [15]. TGF-β1-mediated non-Smad-dependent pathway, also known as the mitogen-activated protein kinase (MAPK) signaling pathway, is involved in cell proliferation, differentiation, transformation, apoptosis, etc. Currently, four MAPK pathways have been discovered, namely extracellular signal-regulated Kinase (ERK), Jun aminoterminal kinases (JNK), ERK5 and p38MAPK pathways [16]. Studies have shown that the TGF-β1-mediated activation of the p38MAPK signaling pathway contributes to the formation of PF [17, 18]. Therefore, TGF-β signaling pathways are considered to be key targets of potential anti-pulmonary fibrosis drugs.

Geniposide (Gen) is a major component mainly derived from the dry and mature fruits of the Gardenia (Gardenia jasminoides Ellis) of the Rubiaceae plant and is an iridoid glycoside

compound. As reported in many studies, Geniposide (methyl (1S,4aS,7aS)-1-(β - D -glucopyranosyloxy)-7-(hydroxymethyl)-1,4a,5,7a- tetrahyd-rocyclopenta[c]pyran-4-carboxylate; C 17 H 24 O 10) has a variety of pharmacological activities [19], including liver protection and choleretic, anti-inflammatory [20], and antioxidant activities. More and more evidence shows that Gen has significant effects in the treatment of inflammation, including excellent performance in acute injury [21], chronic inflammation [22], and reducing macrophage activation [23]. In addition, there are reports that Gen can exert its effects by inhibiting the non-smad-dependent pathway (P38/MAPK) in the TGF-β signaling pathway [24].

As mentioned earlier, the occurrence and development of PF are closely related to early massive pulmonary inflammation and the TGF-β signaling pathway. Our previous research found that Geniopicoside, which also has anti-PF effects [25], has a similar structure to Gen and belongs to the class of iridoid glycosides. However, Gen has a more stable structure and is easier to obtain in nature than Geniopicoside. Based on these, we hypothesized that Geniposide may have a potential therapeutic effect on PF. In the present study, we investigated the protective effects of Geniposide against bleomycin-induced pulmonary fibrosis in mice and explored its possible mechanism of action.

## 2. Materials and methods

### 2.1 Plant material

The dry fruits of Gardenia jasminoides Ellis were purchased from Yunnan Yanshoutang Traditional Chinese Medicine Technology Co., Ltd in May 2019. The plant was identified by Professor Hua Peng, a senior botanist in Kunming Institute of Botany, Chinese Acadamy of Sciences. A voucher specimen (ZC201905) was deposited in the School of Pharmaceutical Sciences & Yunnan Key Laboratory of Pharmacology for Natural Products, Kunming Medical University, China.

### 2.2 Preparation of Geniposide

Ten kg dry fruits of *Gardenia jasminoides* Ellis were powdered and extracted with 95% ethanol 3 times. Every time extracted for 12 hours and filtered. The ethanol extracts were combined and isolated through alumina CC and the ethanol was removed in a vacuum and the resin was isolated through silica gel CC with chloroform-ethanol (9:1), and chloroform-ethanol(8:2) as eluent. Eluent chloroform-ethanol afforded about 500 grams of Geniposide. The purity of Geniposide was >97% as tested by HPLC (S2 Fig).

### 2.3 Animal treatment and experimental programs

A total of 196 Kunming strain adult male SPF mice (7 weeks age and weight 20±2 g; Experimental Animals Center of Kunming Medical University, China) were randomly assigned to 7 groups (n = 28): (1)Single intratracheal instillation of normal saline(NS group); (2) Single intratracheal instillation of bleomycin (Hisun Pfizer, China) 5mg•kg-1body weight(Model group); (3)pirfenidone (Dalian Meilun Biotechnology Co., Ltd., China) 50mg•kg-1body weigh (i.g.) plus treatment in(2)(Positive control group, PFD group);(4) dexamethasone sodium phosphate(Southwest pharmaceutical co., LTD, China) 2.5 mg•kg-1 weight (i.p.)plus treatment in (2)(positive control group, group DXM); (5)High dose of Geniposide 100 mg•kg-1 weight(i.p.) plus treatment in(2)(group Gen100);(6)Medium dose of Geniposide 50 mg•kg-1 weight(i.p.) plus treatment in(2)(group Gen50); (7)Low dose of Geniposide 25 mg•kg-1 weight (i.p.) plus treatment in (2)(group Gen25). On the first day of the experiment, anesthesia was administered by intraperitoneal injection of pentobarbital sodium solution at a dose of 50mg/kg. After 5 minutes of administration, the anesthesia effect was observed and surgery began.

The surgical process should be controlled within 10 minutes. If the anesthesia effect is not good, 1/5 of the original dose should be added immediately. All mice were intratracheally instilled with 40 μL of normal saline (NS group) or bleomycin dissolved in normal saline; then the positive drug PFD (ig), DXM (ip) or different doses of Geniposide (ip) was given, once a day for 28 days. On the 14th and 28th day, 14 mice in each group were anesthetized with a deadly dose of pentobarbital sodium(120mg/kg) respectively, 7 of which were used for collecting bronchoalveolar lavage fluid (BALF), another 7 were used for lung histopathological observation, hydroxyproline (Hyp) content determination and Western blotting (S1 Fig). All experiments were performed following the Declaration of Helsinki and approved by the Ethics Committee of Kunming Medical University (protocol code KMMU2020111, April 2020). All surgery was performed under sodium pentobarbital anesthesia, and all efforts were made to minimize suffering.

## 2.4 Body weight and lung coefficient

From the first day of intratracheal instillation, the body weights of the mice were weighed once on the last day of weeks 0, 1, 2, 3, and 4, and were recorded in the table. When the mice were treated, the lung tissue without alveolar lavage was taken out from the chest of the mouse, and the weight of the lung tissue without the trachea was quickly weighed with a precise electronic balance; the lung coefficient was calculated [lung coefficient (%) = total lung wet weight (mg) / body weight (g)] to reflect the extent of pulmonary edema [26, 27].

## 2.5 Histopathology

The lung tissues from the upper left lobe were fixed in 10% neutral formaldehyde for 24 h, then subjected to conventional tissue block dehydration, paraffin embedding, sectioning, hematoxylin-eosin (H&E) and Masson trichrome staining, the Research-grade upright phase-contrast microscope (Nikon Corporation, Japan) was used for observation and photographing (x100). According to the histopathological scoring method of Szapield [28] and Fulmer et al [29] (S1 File), alveolitis and fibrosis were classified into 4 grades [27]:grade 0, normal (-); grade 1, mild lesion (+); Grade 2, moderate lesion (++); grade 3, severe lesion (+++). The grade data were represented by a scoring method, with level 0 being 1 point, level 1 being 2 points, level 2 being 3 points, and level 3 being 4 points. Three fields of view of each pathological section were randomly selected for scoring and the average score was calculated. Tissue staining and microscopic morphological observation and pathological scoring were performed by professional technicians and experienced pathologists, respectively.

## 2.6 Total cell count and classification count in BALF

Seven mice in each group were euthanized and BALF was obtained as reported [30]. Briefly, after anesthetizing with a lethal dose of pentobarbital sodium, the neck skin was quickly dissected and the mouse trachea was punctured with a 19G disposable indwelling needle; Mouse lung tissue was repeatedly and thoroughly irrigated with pre-cooled normal saline three times, each irrigation spends the 30s and with a volume of 0.4 mL, and after each irrigation, more than 85% of BALF was recovered and centrifuged on high-speed cryogenic centrifuge (4°C, 3000g, 10min), and the supernatant was transferred to a new tube and stored at -80°C, the pelleted cells after centrifugation were resuspended in 100 μL normal saline.

More than 10 μL of the fully suspended cell suspension were collected, and the Count star automatic cell counter (Shanghai Ruihao Biotechnology Co., Ltd., China) was used to perform total cell counting, and then the remaining cell suspension was used for cell smear. After natural drying, conventional dehydration and H&E staining were performed and observed under a

microscope, and a place where cells were uniformly dispersed was photographed (x100). We know that the characteristics of macrophages are large cell volume, abundant cytoplasm, occasional small particles and vacuoles in the cytoplasm, and irregular contours of the nucleus (usually renal type). Granulocytes are characterized by small cell volume, usually with a lobed nucleus (more than 2 lobes visible), and small granules visible in the cytoplasm. The characteristics of lymphocytes are small cell volume and round or oval shaped nuclei; The nucleus occupies most of the cytoplasm, resulting in a low cytoplasmic content and no granules [31]. According to the cell morphology, three inflammatory cells, including macrophages (M), granulocytes (G), and lymphocytes (L), were classified and counted.

## 2.7 Detection of TNF-α and macrophage inflammatory protein-1 (MIP-1α) in BALF

The BALF stored at -80˚C was thawed, the BALF protein concentration was determined by using the BCA method [32], and the sample was diluted according to the assay concentration recommended by the kit manufacturer to make the diluted BALF as the sample to be tested; the ELISA kit (Invitrogen, Canada) was used to determine the levels of TNF-α and MIP-1α.

## 2.8 Hydroxyproline(Hyp) content in lung tissue

Fifty mg of the lung tissue sample from the left lower lobe was hydrolyzed by alkaline hydrolysis, and the content of Hyp was calculated according to the Hyp kit (Nanjing Jiancheng Bioengineering Institute, China) Instruction manual [25].

## 2.9 Analysis of protein expression in lung tissues by western blot

Western blot was used to detect the expression of TGF-β1, Smad2/3, CTGF, and p38 in the lung tissues of mice. The right lung was placed in a pre-cooled tissue homogenizer, incubated with the lysis buffer containing the protease inhibitor, and homogenized well in an ice bath; After centrifugation (8000 g, 10 min) at 4˚C, the supernatant was carefully taken, and the protein concentration of the supernatant was determined by the BCA method. Equal amounts of protein was mixed with the loading buffer, denatured by heating at 95˚C for 10 min, separated by 12% SDS-PAGE electrophoresis gel (20 μg/well), and then transferred to the PVDF membrane (0.40 μm); Membranes were then incubated with primary antibodies against TGF-β1 (1:1000; proteintech, USA), Smad 2/3 (1:1000; CST, USA), CTGF (1:500; R&D Systems, USA) and p38 (1:1000; CST, USA) or GAPDH (1:10000; ProteinTech, USA) for 12 h at 4˚ C. Membranes were then incubated with HRP-conjugated goat anti-mouse IgG pAb (1:5000; ProteinTech, USA). Immunodetection was performed and quantified using the ECL detection system (Bio-Rad, USA), and protein bands were detected by the GE Healthcare Bio-sciences AB imaging system (S1 Raw images and S1 Data).

## 2.10 Molecular docking

In order to verify the binding pattern between Geniposide and the core targets TGF-β1, Smad2, Smad3, and p38 in pulmonary fibrosis, the PDB database was used (https://www.rcsb.org/)to aquire the 3D structure of TGF-β1, Smad2, Smad3, and p38, and PubChem was used to aquire the 3D structure of Geniposide. AutoDockTools 1.5.6 was used for processing Geniposide and using it as a ligand after removing water molecules, adding hydrogen and charges. All small molecules and proteins were converted to pdbqt format. Then autodocktools was used to select a semi-flexible docking method for the ligand and receptor to set the box parameters. The x, y, and z parameters were set until the cube in the graphic completely covers the entire protein and ligand. The parameters were as follows: TGF-β1(grid center:

x = 12.579, y = 1.145, z = 12.494;grid box size: x = 60.0 Å, y = 64.0 Å, z = 60.0 Å); Smad2(grid center: x = 19.229, y = 105.428, z = 25.557; grid box size: x = 78.0 Å, y = 54.0 Å, z = 84.0 Å); Smad3(grid center: x = 9.819, y = 48.232, z = 70.0;grid box size: x = 48.0 Å, y = 50.0 Å, z = 62.0 Å); p38(grid center: x = 21.587, y = 19.964, z = 28.082; grid box size: x = 54.0 Å, y = 42.0 Å, z = 90.0 Å). AutoDock Vina was used to select the conformation with the lowest docking binding energy for docking binding mode analysis. Finally, PyMOL software was used to generate the binding conformation diagram of the ligand and receptor. The binding energy score was used to evaluate the binding specificity of the drug to the target proteins. If the binding energy score is over 0, it means the drug can't bind to the target protein, if the binding energy score is less than 0, the drug can freely bind to the target protein. If the binding energy is less than -5, it means that the drug has good binding activity to the target protein.

### 2.11 In vitro experiment

The cytotoxicity and effect of Geniposide on the activation of macrophages was evaluated in vitro. Raw 264.7 murine macrophage cells(Wuhan Pricella Life Technology Co., Ltd.) were cultured in a humidified atmosphere with 5% $CO_2$ at 37˚C using DMEM-High glucose growth medium supplemented with 10% FBS. Additionally, the culture medium contained penicillin (100 U/mL) and streptomycin (100 µg/mL). Cell Counting Kit-8 (CCK-8) assay was used to evaluate the cytotoxicity of Geniposide. The cells were seeded in a 96-well plate at $1\times10^5$ cells/ mL density. After incubating for 24 hours, different concentrations(0–400µM) of Geniposide were added to the cell cultures. Subsequently, after an additional 24-hour incubation, 10 µL of the CCK-8 reagent was added to each well. The plated samples were then incubated in a 37˚C environment for 2 hours. Following this incubation period, we quantified the absorbance at 450 nm using a Scientific Multiskan GO microplate reader. A specific mathematical formula was applied to calculate the extent of cell proliferation inhibition.

Morphology observation and ELISA assay were used to evaluate the effect of Geniposide on the activation of macrophages. RAW 264.7 cells ($1\times10^5$ cells/mL) were seeded in 6-well plates, with 2 mL per well. After 24 hours, they were treated with LPS (1 µg/ml) + Geniposide (100 µM), LPS (1 µg/ml) + DEX (100 µM). Subsequently, after an additional 24 hours, the morphological changes were observed under the microscope, and the supernatants were collected for the detection of inflammatory cytokines by ELISA assay. For the analysis of TNF-α and IL-1β concentrations in the cell supernatants, we utilized a mouse TNF-α, IL-1β enzyme-linked immunosorbent assay (ELISA) kit obtained from Protientech USA, following the manufacturer's provided instructions.

### 2.12 Statistical analysis

All data were expressed as mean ± SD(standard deviation). The SPSS 20.0 statistical software package was used to test the normal distribution of the data (variance homogeneity test). When the variance was homogeneous, the statistical difference was determined by one-way analysis of variance (ANOVA) and LSD method; the rank sum test was used when the variance was not uniform (Kruskal -Wallis H) Analysis. $P < 0.05$ was considered to be statistically significant.

## 3. Results

### 3.1 Body weight and lung coefficient

At the same stage, the lung coefficient of the model group was significantly higher than that of the NS group and the Gen100 group (P<0.01), but there was no significant difference between the Model group and the PFD group (P>0.05). Interestingly, both PFD and DXM were used

as positive controls, but the lung coefficient of the former was significantly lower than the latter ($P < 0.05$) (Fig 1A and 1B) [$F_A(6,26) = 9.072$, $p = 0.000$], [$F_B(6,34) = 9.238$, $p = 0.000$].

On the 28th day after modeling, there was no significant difference between the Model group and the remaining groups except the NS and DXM groups ($P > 0.05$). At this time, there was no statistical difference between the NS group and the Gen100 group ($P > 0.05$) [$F_A(6,76) = 15.137$, $p = 0.000$]. The growth curve of mice showed that with time, the NS group had the fastest weight gain, while the DXM group had the slowest growth. The growth trends of other groups were similar. In the other groups except for NS and DXM, the weight gain of mice in the PFD group was relatively slower, presumably due to the gastrointestinal adverse reactions of PFD caused by long-term gavage (Fig 1C).

## 3.2 Pathological morphology of mouse lung tissue

The effects of Gen on BLM-induced mouse alveolitis and interstitial fibrosis were assessed by light microscopy of H&E and Masson staining, and pathological differences were described

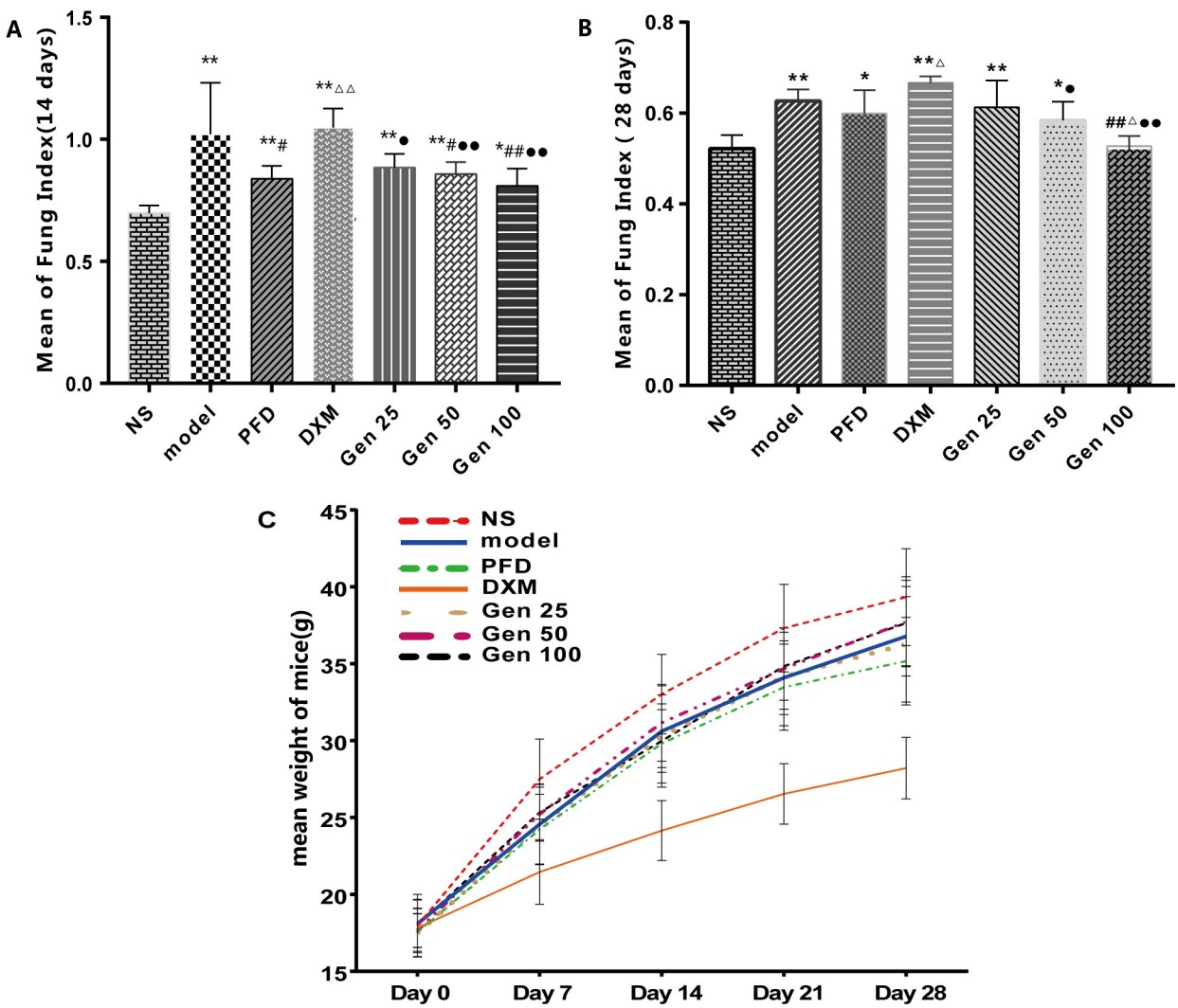

**Fig 1. Geniposide (Gen) significantly improved lung coefficient and growth curve in bleomycin (BLM)-induced pulmonary fibrosis in mice.** As shown in the figure, (A,B) is the lung coefficient of mice on day 14 and 28 (lung coefficient = lung wet weight/body weight x 100%). (C) The growth status (weight, g) of mice from day 1 to day 28 of the experiment. Compared with group NS, model and PFD,* $P < 0.05$, ** $P < 0.01$; # $P < 0.05$, ## $P < 0.01$; and △$P < 0.05$, △△$P < 0.01$; DMX: •$P < 0.05$,••$P < 0.01$.

using a scoring system of alveolitis and fibrosis. The staining results showed that the model group had a large amount of inflammatory cell infiltration, thickened alveolar wall and widened alveolar space compared with the control group. A small amount of alveolar collapse, alveolar septal rupture, and lymphocytic infiltration were observed on the 28th day, and a large amount of collagen deposition was observed around the pulmonary interstitial and bronchi, and collagen was still accumulated on the 28th day (Fig 2①, 2②). Gen can significantly improve pathological changes including alveolitis and fibrosis, especially on the 28th day. However, it is worth noting that although DXM can effectively improve alveolitis, it does not improve collagen accumulation in the lung interstitial and around the bronchi (Fig 2).

Histopathological scores showed a significant increase in alveolitis and interstitial fibrosis scores in the model group compared with NS (P < 0.01), and the alveolitis score peaked on the 14th day and then decreased, but the interstitial fibrosis score continuously increased and reached the peak on the 28th day. Compared with the model group in the same period, Gen treatment significantly reduced the increase of alveolitis and interstitial fibrosis scores (P<0.01), and its effect was equivalent to PFD (P>0.05); but unfortunately, DXM did not reverse the increase in fibrosis score (P>0.05) (Table 1).

### 3.3 Total cell count and differential count in bronchoalveolar lavage fluid (BALF)

The morphology of cells in bronchoalveolar lavage fluid of mice in different groups is shown in Fig 3A. Compared with the NS group, the total number of cells in the BALF of the model group was significantly increased (P<0.05) [$F_B(6,13) = 3.884$, p = 0.019], and gradually increased with time. At the same stage, compared with the model group, Gen100 significantly reduced the total number of cells in BALF (P<0.05), and the effect was similar to PFD and DXM (P>0.05) [$F_C(6,27) = 9.856$, p = 0.000] (Fig 3B and 3C). The classification of inflammatory cells showed that BLM significantly increased the content of three types of cells (M/G/L) in BALF in different stages (P<0.01), while Gen100 significantly reduced the number of the three types of inflammatory cells in BALF(P < 0.01). Dexamethasone appeared to have a more pronounced inhibitory effect on granulocytes as a glucocorticoid and was superior to PFD (P<0.01)[$F_M(6,33) = 32.249$, p = 0.000],[$F_G(6,29) = 20.771$, p = 0.000], $F_L(6,20) = 8.708$, p = 0.000]. It is worth noting that on the 14th day, the inflammatory cells were mainly macrophages, but on the 28th day, lymphocytes that characterizes chronic inflammation dominated in inflammatory cells (Fig 3D and 3E).

The cells obtained after centrifugation of BALF were subjected to H&E staining and photographed at a position where the cells were uniformly dispersed (x200/3 sheets). Based on the morphological characteristics of macrophage (M), granulocyte (G) and lymphocyte (L) nuclei, the number of three types of inflammatory cells (M/G/L) in each photograph was counted and the average was calculated (Set to $\bar{b}_{M/G/L}$). Count the total cells and three types of cells (M/G/L) on each photo and calculate the average value (set to $\bar{a}$ and $\bar{b}_{M/G/L}$), which is regarded as the count value of one sample. Cell proportion of $= \bar{b}_{M/G/L} / \bar{a}$. Compared with group NS, model and PFD,* P<0.05, ** P<0.01; # P<0.05, ## P<0.01; and △P<0.05, △△P<0.01; DMX: •P<0.05,••P<0.01.

### 3.4 Content of TNF-α and MIP-1α in BALF

Compared with the NS group, the expression of TNF-α and MIP-1α in the model group induced by BLM was significantly increased and peaked on the 14th day (P<0.01) [$F_A(6,25) = 7.087$, p = 0.000],[$F_C(6,21) = 7.261$, p = 0.000] (Fig 4A and 4C). Compared with the model

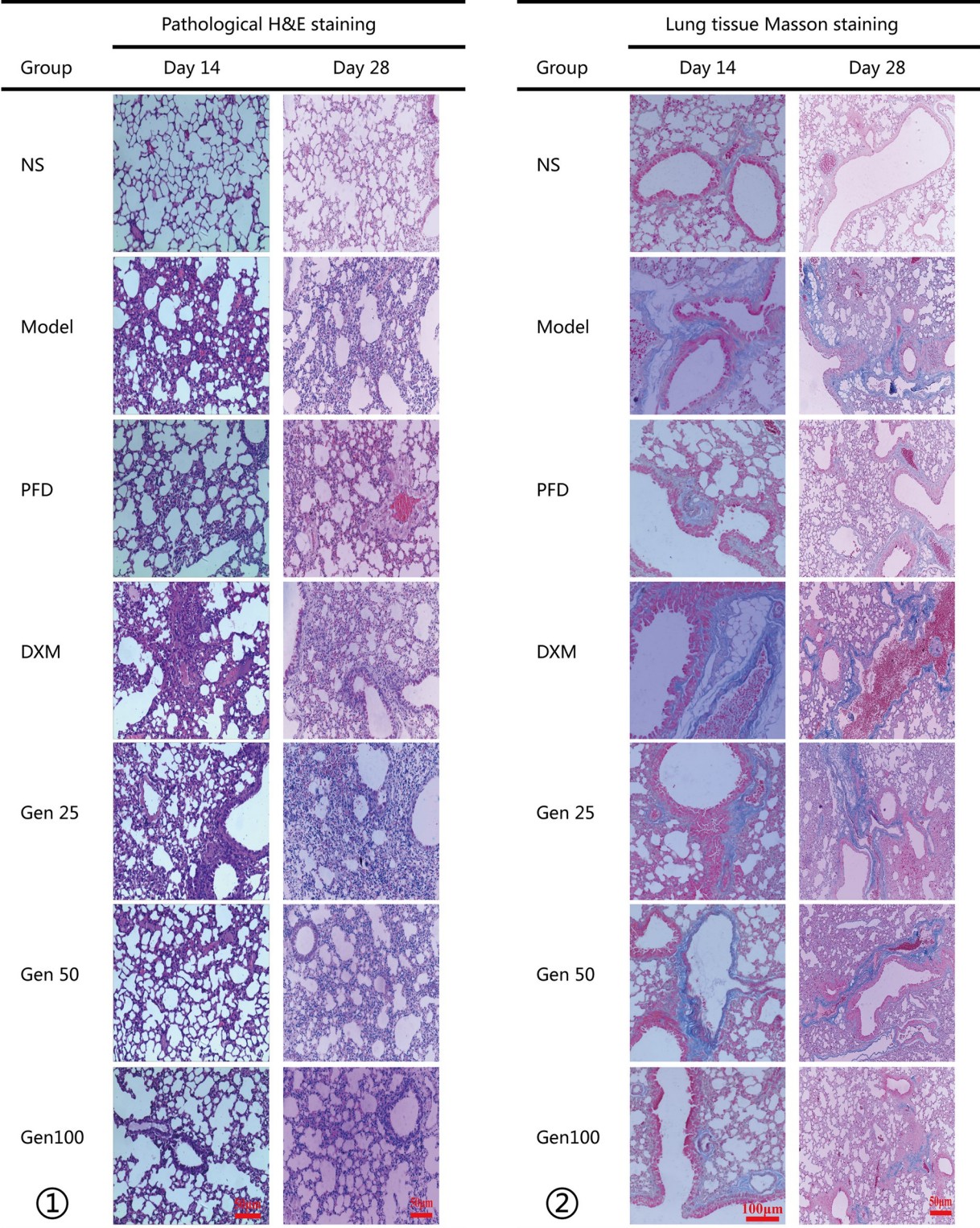

**Fig 2. Geniposide (Gen) significantly ameliorates bleomycin (BLM) -induced inflammatory and fibrotic lesions in the lungs.** (Fig 2-①) Representative image of H&E staining in lung tissue (scale bar, 100 μm). (Fig 2-②) Representative image of Masson staining in lung tissue (scale bar, 100 μm); The blue area in Fig 2-② represents collagen deposition.

**Table 1. Geniposide (Gen) significantly reduced histopathological scores in mice with pulmonary fibrosis induced by bleomycin.**

| Group | Alveolitis score(H&E staining) | | Pulmonary interstitial fibrosis score(Masson staining) | |
|---|---|---|---|---|
| | Day14 | Day 28 | Day14 | Day 28 |
| NS | 1.80±0.45 | 1.29±0.39 | 1.17±0.26 | 1.00±0.00 |
| Model | 3.3±0.27$^{**}$ | 2.92±0.58$^{**}$ | 2.90±0.55$^{**}$ | 3.17±0.26$^{**}$ |
| PFD | 2.40±0.22$^{*##}$ | 1.75±0.27$^{*##}$ | 1.58±0.38$^{##}$ | 1.43±0.45$^{##}$ |
| DXM | 2.58±0.49$^{**##}$ | 2.13±0.25$^{**##}$ | 2.88±0.25$^{**△△}$ | 3.13±0.48$^{**△△}$ |
| Gen 25 | 2.88±0.25$^{**}$ | 2.60±0.42$^{**△△}$ | 2.60±0.22$^{**△△}$ | 2.88±0.25$^{**△△}$ |
| Gen 50 | 2.50±0.41$^{**##}$ | 2.17±0.26$^{**##}$ | 2.17±0.68$^{**#△•}$ | 2.17±0.41$^{**##△△••}$ |
| Gen 100 | 2.38±0.25$^{*##}$ | 1.71±0.39$^{*##}$ | 1.83±0.52$^{*##••}$ | 1.70±0.67$^{*##••}$ |

Data are expressed as mean ± SD; Compared with group NS, model and PFD

* P<0.05

** P<0.01

# P<0.05

## P<0.01; and

△P<0.05

△△P<0.01; DMX

•P<0.05

••P<0.01.

group, Gen significantly reduced the levels of TNF-α and MIP-1α in different periods (P<0.05) and its effect was similar to PFD (P>0.05) (Fig 4). It is worth noting that the content of two cytokines in the DXM group was higher than that in the PFD group on the 28th day (P<0.05), and the content of MIP-1α was even close to that of the model group (P>0.05) [$F_B(6,20)$ = 5.997, p = 0.001],[$F_D(6,21)$ = 11.841, p = 0.000] (Fig 4B and 4D). These data suggest that the powerful anti-inflammatory advantage of dexamethasone appears to be gradually lost in chronic inflammation.

## 3.5 Hyp content in lung tissue

BLM significantly increased the content of Hyp in the lung tissue of the model group compared with the NS group (P<0.01), and peaked on the 28th day. Compared with the model group, Gen100 significantly reduced the Hyp content (P<0.01) and was similar to PFD (P>0.05) [$F_A(6,24)$ = 10.166, p = 0.000]. In addition, Gen's effect on decreasing the content of Hyp showed a dose-dependent manner. We noted that PFD, which was newly approved for the treatment of PF, showed excellent results (P < 0.01), but DXM with anti-inflammatory advantages did not significantly decrease the level of Hyp in lung tissue (P > 0.05) [$F_B(6,16)$ = 7.386, p = 0.001] (Fig 5).

## 3.6 Expression of TGF-β1 /Smad2/3 protein in lung tissue

The relative expression levels of TGF-β1 and Smad2/3 in the model group were significantly increased compared with the NS and PFD groups (P<0.01). Compared with the model group, Gen100 significantly reduced the relative expression of TGF-β1 and Smad2/3 in lung tissue (P<0.01) and was comparable to the PFD group (P>0.05) [$F_B(6,24)$ = 6.967, p = 0.000], [$F_C(6,29)$ = 8.178,p = 0.000] (Fig 6). Interestingly, the Gen25 group only reduced the relative expression of Sand2/3 on the 28th day (P<0.05), and the rest were not different from the model group (P>0.05) [$F_E(6,23)$ = 5.963, p = 0.001], [$F_F(6,24)$ = 19.275, p = 0.000]. The results indicate that Gen downregulates the expression of TGF-β1 and Smad2/3 in a dose-dependent manner. In addition, there was no statistically significant difference in the expression of TGF-

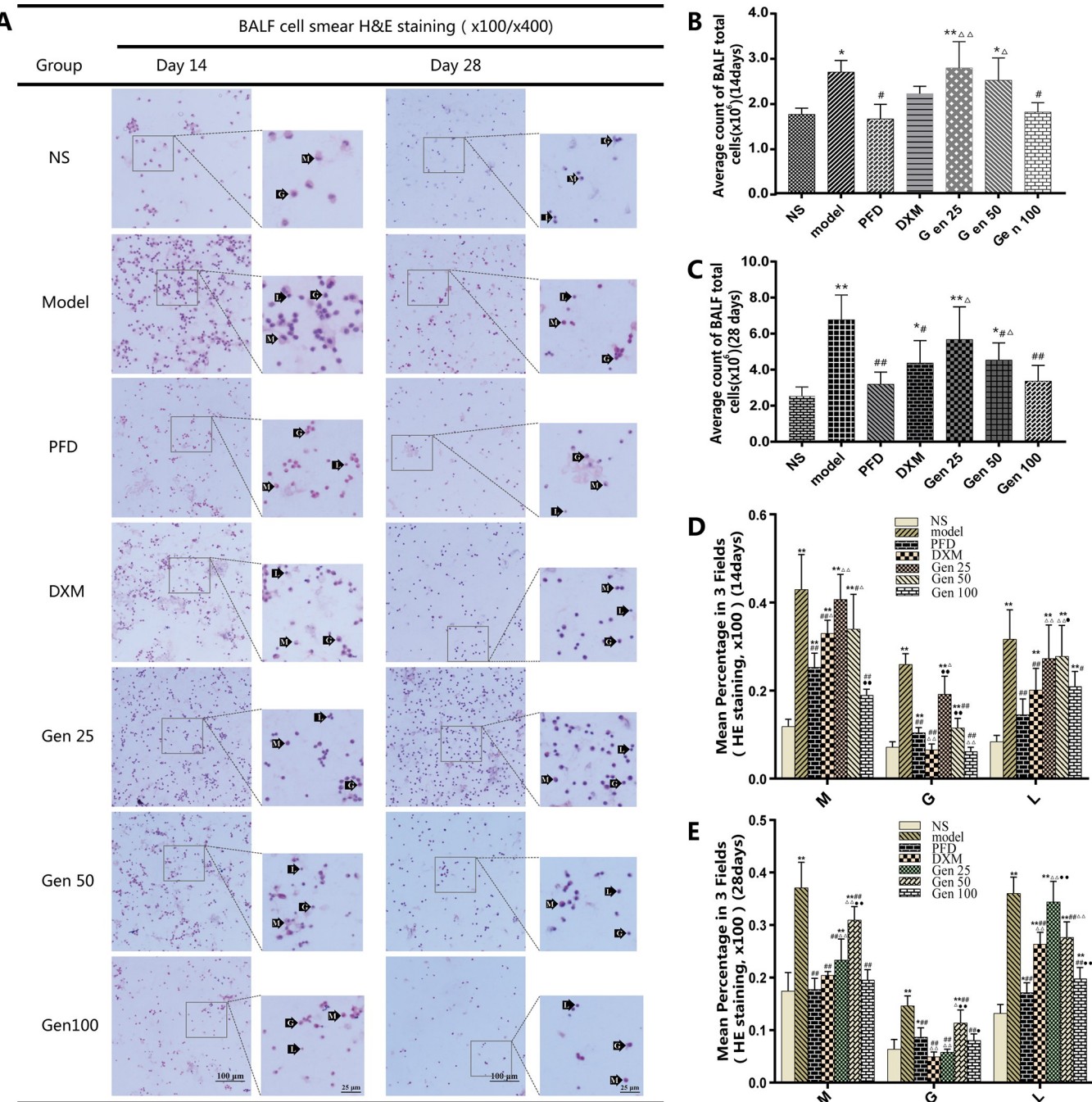

**Fig 3. Geniposide (Gen) significantly reduces the number of inflammatory cells in BALF in PF mice.** (A)Representative images of cell smears stained in BALF. (B-C)Quantitative analysis of the total number of inflammatory cells in BALF. (D-E)Quantitative analysis of three types of inflammatory cells.

β1 on the 28th day between the DXM group and model group (P>0.05) (Fig 6E), indicating that DXM could not effectively inhibit the expression of TGF-β1 in the lung tissue.

## 3.7 Expression of CTGF and p38 in lung tissue

On the 14th and 28th day after BLM induction, the expression levels of CTGF and p38 protein in the lung tissue of mice in the model group were significantly higher than those in the NS

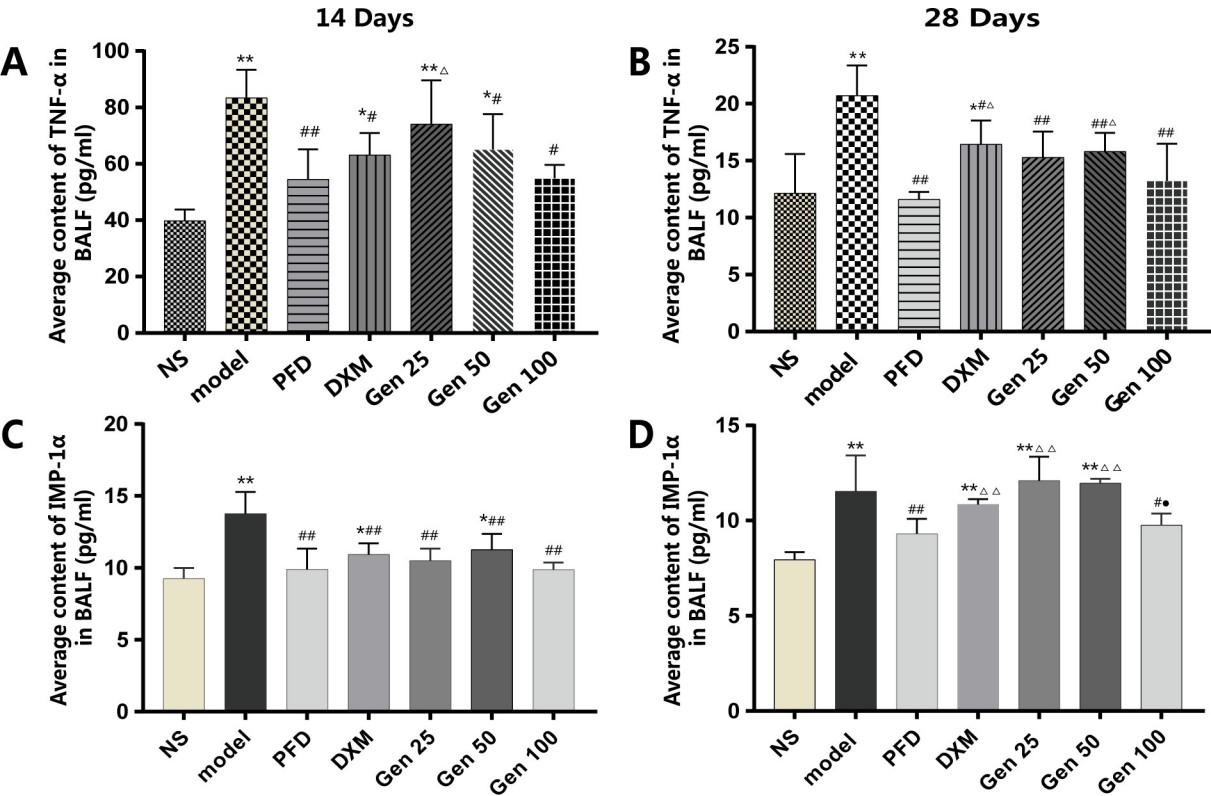

**Fig 4. Geniposide(Gen) significantly decreased the levels of TNF-α and MIP-1α in BALF of bleomycin-induced PF mice.** NS, model, PFD, DXM, Gen25/50/100 group are the same as in Fig 2. The expression of inflammatory cytokines TNF-α and MIP-1α at different time points were examined by ELISA kit to verify the effect of Geniposide on inflammatory factors at different stages of pulmonary fibrosis. Compared with group NS, model and PFD,* P<0.05, ** P<0.01; # P<0.05, ## P<0.01; and △P<0.05, △△P<0.01; DMX: •P<0.05.

group (P<0.01) [$F_B$(6,25) = 9.466, p = 0.000], [$F_C$(6,28) = 12.001,p = 0.000]. Compared with the model group, the expression levels of CTGF and p38 protein on the 14th and 28th day after BLM induction in the Gen100 group were significantly lower (P<0.01), the expression levels of CTGF on the 14th and 28th day, and p38 on the 28th day after BLM induction in the Gen50 group were significantly lower(P<0.05), but only the expression of p38 on the 28th day in the Gen 25 group was significantly decreased (P<0.05) [$F_E$(6,24) = 18.592, p = 0.000], [$F_F$(6,31) = 15.772, p = 0.000]. The results indicate that Geniposide downregulates the expression of CTGF and p38 in the lung tissue of mice with pulmonary fibrosis in a dose-dependent manner (Fig 7).

### 3.8 Molecular docking

Molecular docking was performed between Geniposide and the core targets, and the docking scores are shown in Table 2. As shown in Fig 8, the amino acid residues of the binding active site of Geniposide with TGF-β1 were LEU-278, LYS-232, TYR-249, SER-280, ALA-230, ASP-351, and HIS-283. The amino acid residues of the binding site of Geniposide with Smad2 were THR-298, ARG-310, and ASN-307, and Geniposide could also bind to the amino acid residues GLU-245, SER-263, and SER-265 of the binding site of Smad3. Finally, the hydrogen bond between Geniposide and p38 protein acted on the amino acid residues ASN-155, GLY-154, ASP-113, ASP-168, MET-110, and LYS-54. The binding energy score was used to evaluate the binding specificity, and a score less than -5 was usually a criterion for specific binding. The binding energy scores of TGF-β1, Smad2, Smad3, and p38 were -8.6, -6.2, -6.8 and -6.9,

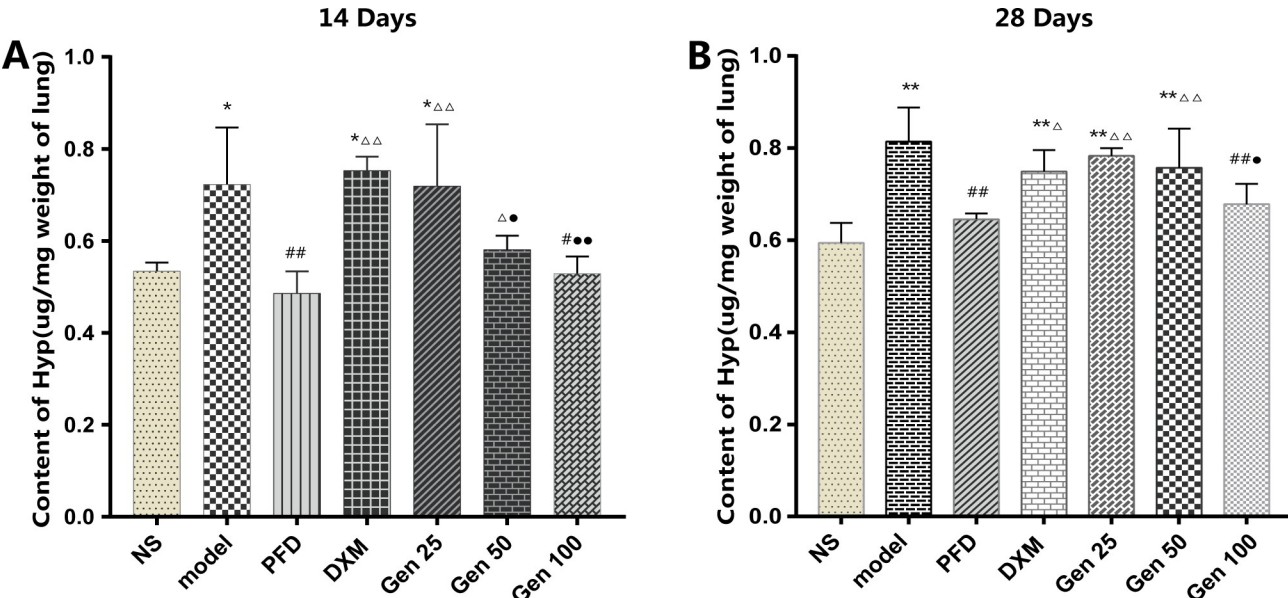

**Fig 5. Geniposide (Gen) significantly decreased the content of Hyp in bleomycin-induced PF mice's lung tissue.** (A-B) represent the content of Hyp in lung tissue on days 14 and 28, respectively. Compared with group NS, model and PFD,* P<0.05, ** P<0.01; # P<0.05, ## P<0.01; and △P<0.05, △△P<0.01; DMX: •P<0.05,••P<0.01.

respectively. These results suggest that TGF-β1, Smad2, Smad3, and p38 have good binding activity with Geniposide.

## 3.9 Effect of Geniposide on the activation of Raw 264.7 cells induced by LPS

The toxicity of Geniposide on Raw 264.7 cells was evaluated, and the results are shown in Fig 9. Geniposide had lower toxicity than dexamethasone on Raw 264.7 cells, with $IC_{50}$ >400 μM

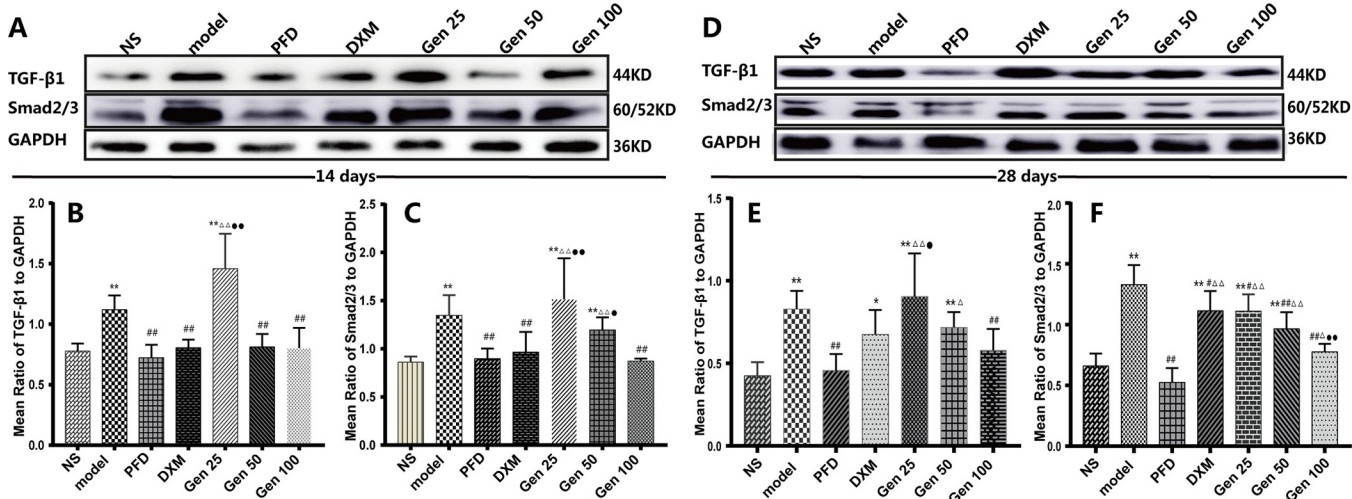

**Fig 6. Geniposide (Gen) significantly down-regulated the expression of TGF-β1and Smad2/3 in lung tissue of bleomycin-induced PF mice.** (A-C) Representative blot images and quantitative analysis of TGF-β1and Smad2/3 on 14th day. (D-F) Representative blot images and quantitative analysis of TGF-β1and Smad2/3 on 28th day. Compared with group NS, model and PFD,* P<0.05, ** P<0.01; # P<0.05, ## P<0.01; and △P<0.05, △△P<0.01; DMX: •P<0.05,••P<0.01.

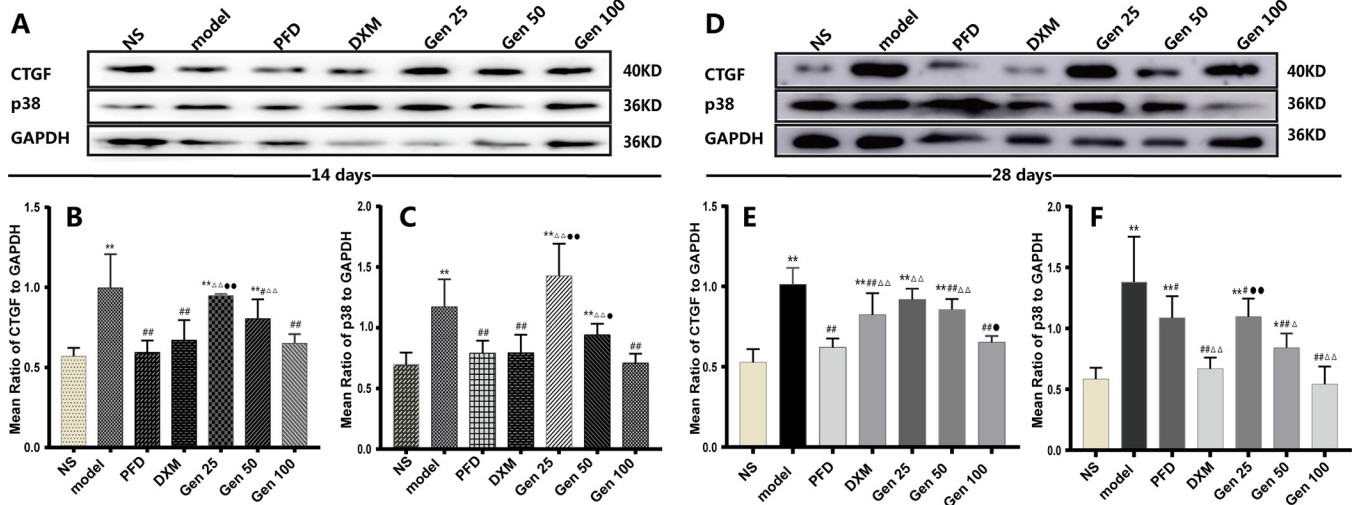

**Fig 7. Geniposide significantly down-regulated the expression of CTGF and p38 in lung tissue of bleomycin-induced PF mice.** (A-C) Representative blot images and quantitative analysis of CTGF and p38 on 14th day. (D-F) Representative blot images and quantitative analysis of CTGF and p38 on 28th day. Compared with group NS, model and PFD,* $P < 0.05$, ** $P < 0.01$; # $P < 0.05$, ## $P < 0.01$; and △$P < 0.05$, △△$P < 0.01$; DMX: •$P < 0.05$,••$P < 0.01$.

and $IC_{50} = 330.169\mu M$, respectively. Geniposide and dexamethasone at $100\mu M$ had no statistically significant influence on the cell viability of Raw 264.7 cells (Fig 9). Therefore, $100\mu M$ was selected for evaluating the effect of Geniposide on the activation of Raw 264.7 cells induced by LPS.

As shown in Fig 10, normal Raw 264.7 cells were round and grew in clusters, consistent with the characteristics of M0 macrophages. After LPS stimulation, the cells increased in size and took on polygonal or irregular shapes with obvious pseudopodia, which are typical characteristics of M1 macrophages. Following treatment with 100 μM Geniposide and dexamethasone, the morphology of LPS-induced Raw 264.7 cells underwent varying degrees of change, cells displayed significant elongation in pseudopodia, with more long spindle-shaped cells, suggesting that geniposide and dexamethasone inhibited the polarization of Raw 264.7 cells to M1 macrophages induced by LPS. Meanwhile, both Geniposide and dexamethasone significantly decreased the contents of M1 cytokines IL-1β and TNF-α in LPS-induced Raw 264.7 cell culture supernatants, suggesting Geniposide could inhibit the release of M1 cytokines of macrophages.

## 4. Discussion

This study explored the therapeutic effect of Gen on PF mice and its possible mechanisms, providing new evidence for Gen's treatment of pulmonary fibrosis. Our research has found that Gen can inhibit the inflammatory response during pulmonary fibrosis and also inhibit TGF- β/Smad and p38MAPK signaling pathways to alleviate bleomycin-induced pulmonary fibrosis in mice.

**Table 2. Molecular docking patterns of Geniposide with core targets in pulmonary fibrosis.**

| Target | PDB | Binding sites with the amino acid | Score (kcal/mol) |
|---|---|---|---|
| TGF-β1 | 3TZM | LEU-278, LYS-232, TYR-249, SER-280, ALA-230, ASP-351, HIS-283 | -8.6 |
| Smad2 | 1DEV | THR-298, ARG-310, ASN-307 | -6.2 |
| Smad3 | 1MK2 | GLU-245, SER-263, SER-265 | -6.8 |
| p38 | 3COI | ASN-155, GLY-154, ASP-113, ASP-168, MET-110, LYS-54 | -6.9 |

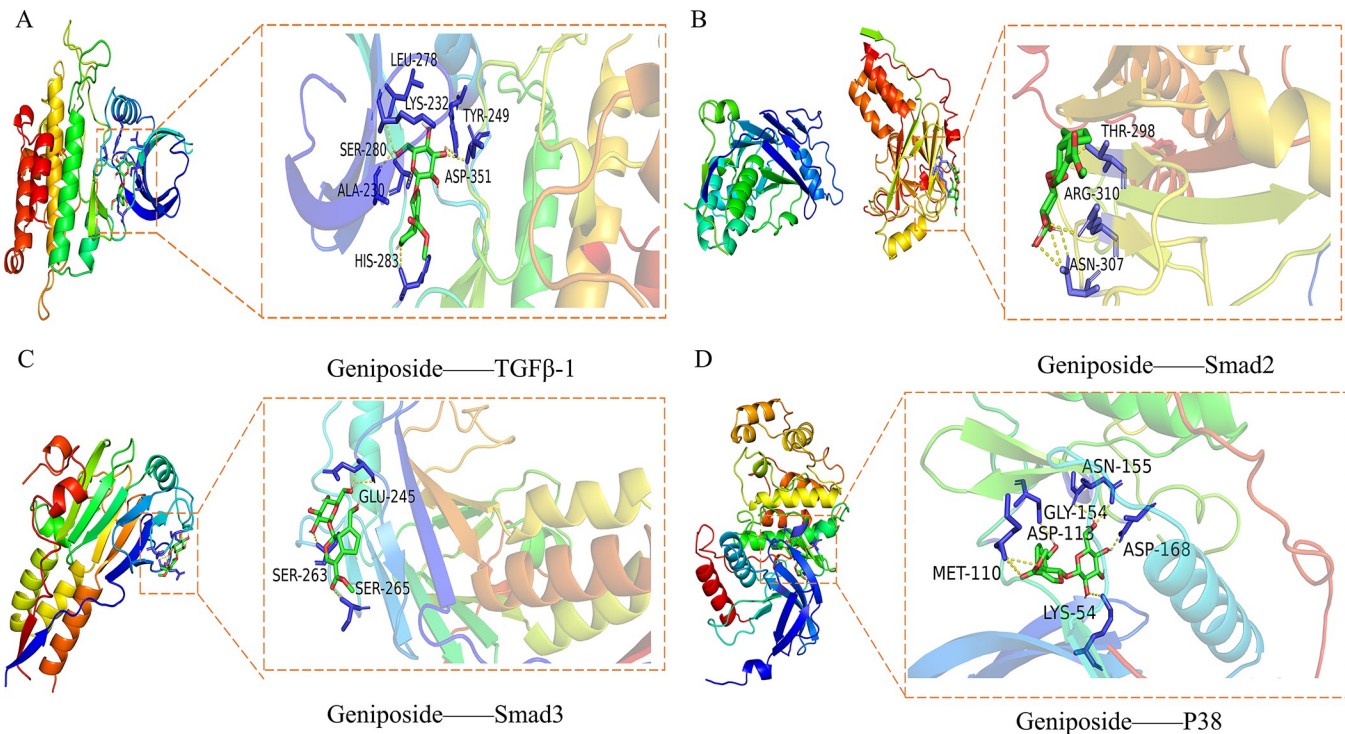

**Fig 8. The docking patterns of Geniposide with core targets in pulmonary fibrosis.** (A)The docking binding site of Geniposide and TGF- β1. (B)The docking binding site of Geniposide and Smad2. (C)The docking binding site of Geniposide and Smad3. (D)The docking binding site of Geniposide and p38.

The pathogenesis of PF is complex and diverse. Most of the understanding and interpretation of PF is only in morphological changes, such as damage of alveolar epithelial cells, the excessive proliferation of fibroblasts, widening of the alveolar septum, massive inflammatory cell infiltration, excessive collagen deposition, and scar formation [33, 34]. BLM-induced fibrosis models can replicate these features of PF [7]. It has been reported that regardless of which route of administration induces PF, BLM-induced lung injury will undergo three processes: acute injury and inflammation, the transition from inflammation to fibrosis, and chronic fibrosis stage [7]. In this study, we carefully considered the selection of gender for model animals. Chioma et al. [35] confirmed that estrogen has a promoting effect on the progression of PF by comparing normal female mice with female mice undergoing oophorectomy. In another study, Elliot et al. [36] investigated the relationship between PF in male mice and the expression of estrogen receptors in lung tissue, and the results showed that reduced expression of estrogen receptors could inhibit the development of PF. Based on the above evidence and to avoid the influence of estrogen on the experimental results, we used male mice in this study.

In the first 7 days after intratracheal instillation, BLM mainly caused an inflammatory reaction and increased apoptosis of epithelial cells [37]; by the 14th day, inflammation of the lungs gradually declined and the fibrosis began to form; thereafter, chronic inflammation and continued development of fibrosis were dominated [7]. In this study, we gave Gen treatment simultaneously from the first day of the intratracheal instillation of BLM. Pathological examination of the lungs revealed a sharp increase in the pathology score of the model group and a large amount of inflammatory cell infiltration on the 14th day; however, by the 28th day, this phenomenon had been converted to lymphocyte infiltration. Interestingly, Gen significantly reduced lung histopathology scores upregulated by BLM and improved this inflammatory

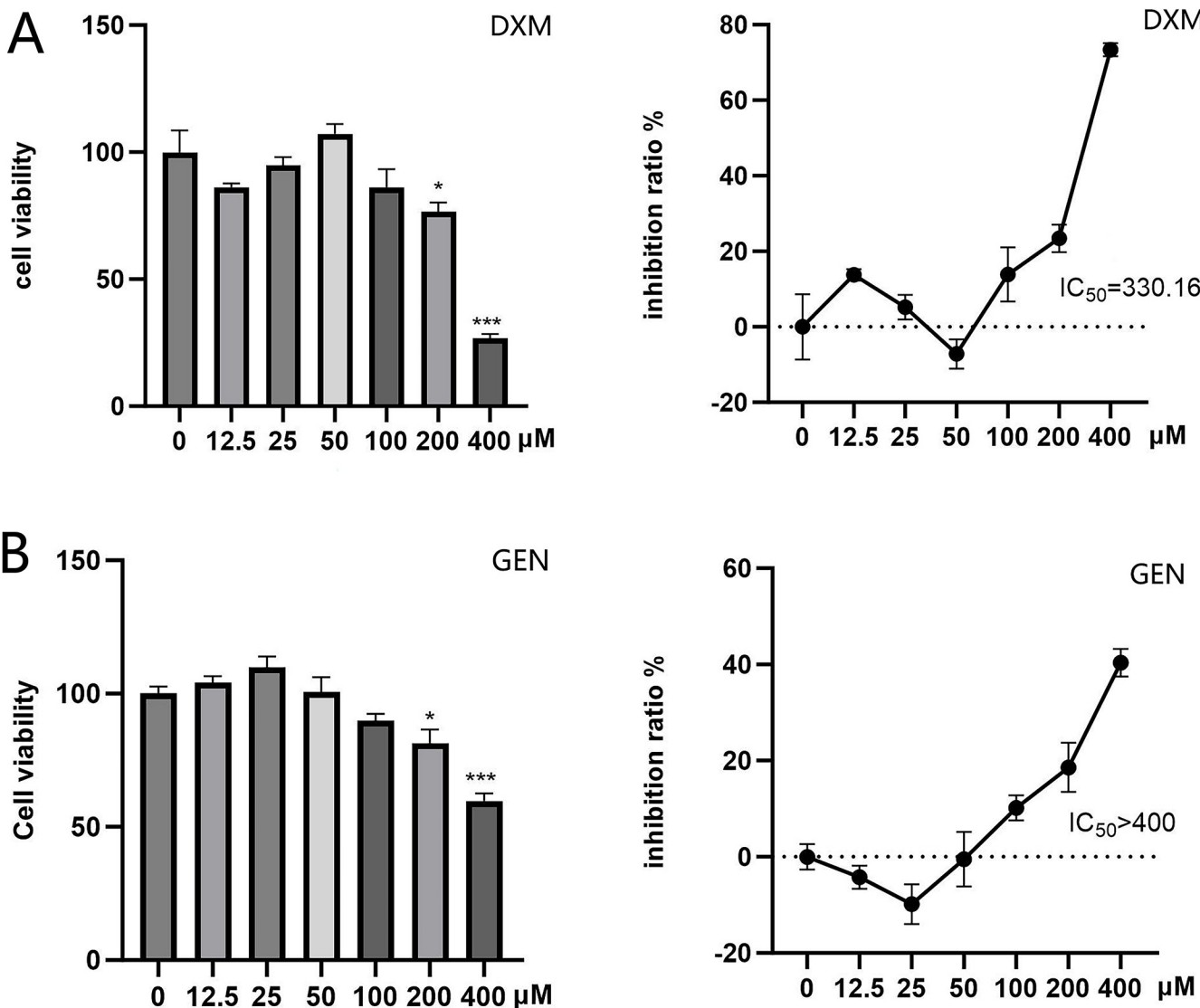

**Fig 9. Effect of Geniposide on cell viability of Raw264.7 cells.** Pretreatment of RAW264.7 cells with GEN(A) and DXM(B) for 24 hours and cell viability was subsequently assessed using the Cell Counting Kit-8 assay. Compared with 0 μM, *P<0.05;**P<0.01.

response. At the same time, our data show that Gen can significantly reduce the increase in the lung coefficient in mice caused by BLM, indicating that Gen attenuates BLM-induced lung injury, which is consistent with the results of lung tissue H&E staining. However, it is noteworthy that the lung coefficient of the DXM group Always remained at a high level throughout the study (28 days); of the mouse growth curve, the DXM group also had a slower weight gain. This suggests that we need to be careful consideration when using glucocorticoids as a positive control in anti-PF drug studies.

To explore the effects of Gen on the inflammatory response during the development of PF, we further analyzed BALF. At present, the use of BALF cell analysis as an indicator in the diagnostic evaluation of suspected PF patients is still controversial. However, after excluding interference factors such as infection and malignant tumors, some experts supported the BALF differential cell count as an adjunct to the clinical diagnosis and evaluation of PF patients [38]. In our study, the total number of cells in BLM-induced mouse BALF was significantly higher

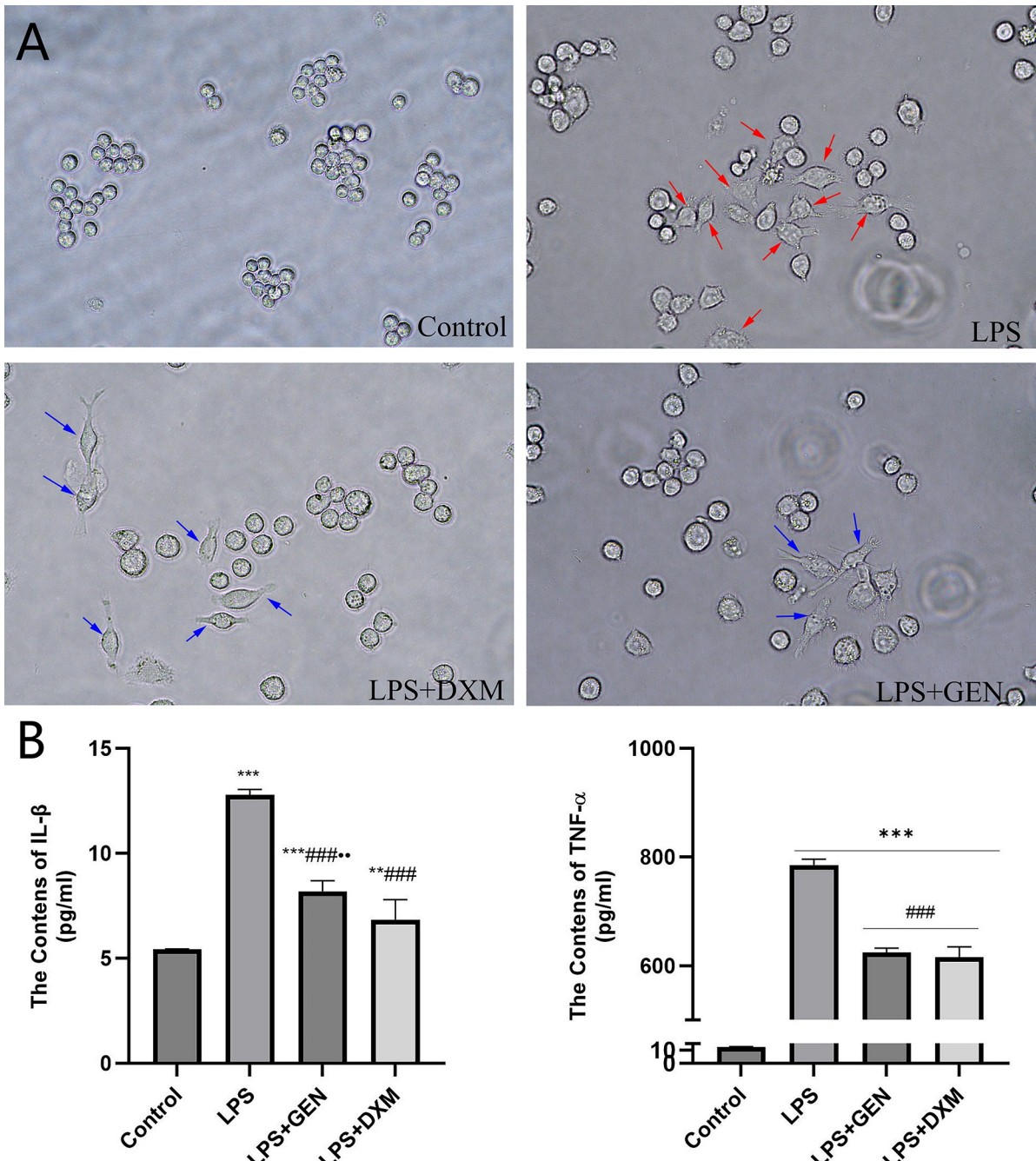

**Fig 10. Geniposide inhibited the activation and release of inflammatory cytokines of Raw264.7 cells induced by LPS.** (A) Effect of geniposide on morphology of LPS-induced Raw264.7 cells, Red arrows show the LPS-activated Raw264.7 cells(polygonal or irregular shapes with obvious pseudopodia); blue arrows show the long spindle-shaped cells with significant elongation in pseudopodia. (B) Effect of geniposide on inflammatory cytokines in LPS-activated Raw264.7 cell culture supernatants. Control: normal Raw264.7 cells; LPS: Raw264.7 cells treated with 1 μg/ml LPS; LPS+DXM: Raw264.7 cells treated with 1 μg/mL LPS and 100 μM dexamethasone(DXM); LPS+GEN: Raw264.7 cells treated with 1 μg/mL LPS and 100 μM geniposide(GEN). Compared with Control group, **P<0.01, ***P<0.001; Compared with LPS group,###P<0.001; Compared with LPS+DXM group, ••P<0.01.

than that in NS mice, which may be the result of BLM causing alveolar epithelial cell damage and apoptosis and induction of a product with a large number of inflammatory cells [39]; Interestingly, Gen significantly reduced the total number of cells in BALF, indicating that Gen can effectively improve lung damage caused by BLM.

Macrophages are a heterogeneous population of resident and recruited cells, characterized by their ability to respond to environmental stimuli and significantly alter their morphology and physiological functions [40]. Activated macrophage characteristics are heralded by a significant increase in cell size, an increase in the number of cytoplasmic granules, an increase in the heterogeneity of cell size and shape, and an increase in the number of cytoplasmic clear vacuoles in comparison to monocytes. The cell morphology exhibits folds, pseudopodia, and the nucleus variations in shape from horseshoe to fusiform [41]. In this study, we classified and counted the M/G/L inflammatory cells, and found that the granulocytes (G) in the BALF of the model group increased sharply on the 14th day, and the G decreased on the 28th day, but the lymphocytes (L) Increased obviously; in addition, we found that macrophages had always been present in high numbers. This was consistent with the pathological examination described above. However, what excited us was that Gen significantly reduced the number of M/G/L in BALF and its inhibitory effect on M/G was better than L. It should be noted that the effect of DXM on L on day 28 was not significant compared with the NS group; this also indicates that DXM is useful for acute inflammation in a short period, but its role in chronic inflammation is not significant.

Macrophage inflammatory protein (MIP) -1α (MIP-1α) plays an important role in the inflammatory response by recruiting monocytes, regulating the production of inflammatory cytokines, and causing fever [42]. Studies have reported that MIP-1α has the property of activating macrophages; it plays a role in regulating the body's response to inflammation-stimulated macrophages, which is characterized by stimulating the proliferation of macrophages and causing macrophage secretion of TNF, IL-1α and IL-6 [43]. We found that MIP-1α and TNF-α in the model group were consistently higher than those in the NS group at the same stage, and the content of both decreased gradually with time, which is consistent with the pathological features of the BLM-induced PF model; Encouragingly, Gen significantly reduced the levels of MIP-1α and TNF-α in BALF, suggesting that Gen is resistant to inflammatory responses during pulmonary fibrosis, which was also confirmed in pathological examinations. Besides, we have learned from previous reports [44, 45] that Gen can also exert anti-inflammatory effects by inhibiting the nuclear factor (NF) -κB/MAPK/activator protein (AP) -1/PI3K/ AKT signaling pathway. In summary, we believe that Gen may slow down the progression of PF by inhibiting the inflammatory response.

The content of Hyp in lung tissue is considered to be a fibrosis marker of deposit collagen, which can indirectly reflect the extent of PF [27]. In our study, the Hyp content of the model group increased significantly on day 14 and peaked on day 28 compared to the NS group, which is consistent with previous reports [7, 25]; however, our The data indicate that Gen reduces the Hyp content at any stage after BLM treatment in a dose-dependent manner (Fig 5), indicating that Gen can reduce the collagen content in lung tissue and reduce collagen deposition during the development of PF. The pathological examination further confirmed the anti-PF effect of Gen, and Gen significantly reduced the level of collagen in Masson-stained lung tissue compared with the model group (Fig 2②). Also, our experimental results show that DXM has a certain inhibitory effect on the content of Hyp and collagen deposition, which is similar to the previous study [46]; but it does not show a significant decrease compared with the NS group (Figs 2② and 5).

It is well known that TGF-β plays an important role in the development of PF, which regulates a variety of cellular functions by binding to and activating specific cell surface receptors

with intrinsic serine/threonine kinase activity [47]. Studies have shown that members of the Smad family are essential components of TGF-β superfamily signaling in cells. Smad2 and Smad3 are structurally highly similar, whereas phosphorylation of Smad4 does not increase with TGF–β stimulation, but It is noteworthy that the accumulation of Smad2/3 and Smad4 in the nucleus is increased after TGF-β stimulation [48], indicating that Smad2/3 mediates TGF-β signaling in cells. Of course, the Smad-dependent pathway of TGF-β is now well understood [47], but the intrinsic link between TGF–β1 and CTGF in the PF process needs to be elucidated. Studies have shown that CTFG is an effector downstream of TGF–β1, and as a functional intermediate between ECM protein and TGF–β1, it is an early gene that is up-regulated in response to TGF–β1 stimulation [15]. The researchers identified Smad binding sites from the mouse fibroblast CTGF promoter and confirmed that Smad3 (but not Smad2) is required for TGF-β1 to induce upregulation of CTGF [49] when Smad2/3 is inhibited It will attenuate the expression of CTGF up-regulated by TGF-β1. However, it should be noted that Smad is not the only pathway by which TGF–β1 induces CTGF expression, which may vary with species and cell types [50]. Besides, the intracellular signaling of TGF-β also has a Non-Smad-dependent pathway [51]. According to reports, as early as 2002, Hiroto Matsuoka et al [52] considered that p38MAPK is involved in BLM-induced PF, and clarified that phosphorylation of p38MAPK continued from day 1 to day 21 after BLM exposure. Moreover, p38 kinase inhibitor (pirfenidone) has been approved for the treatment of PF [53], and subsequent studies have further confirmed the association between TGF-β and p38MAPK in PF [54, 55]. Our study showed that the relative expression of TGF-β1, Smad2/3, CTFG, and p38 proteins was significantly up-regulated in BLM-treated mouse lung tissues, whereas Gen significantly down-regulated their expression in a dose-dependent manner(Figs 6 and 7). Besides, we observed that DXM only down-regulated the p38 protein on day 28; this possible explanation is that p38MAPK is also involved in the inflammatory process [45]. These results indicate that the dysregulation of TGF-β1 /Smad2/3, CTGF, and p38 protein is closely related to the formation of PF, while Gen can significantly reduce their expression and alleviate PF; meanwhile, for chronic progressive PF, DXM seems to not play an effective therapeutic role. The reason may be related to the uncontrolled expression of fibrotic factors such as TGF-β1/Smad2/3 and CTGF. Molecular docking results showed that Geniposide has good binding activity with TGF- β 1, Smad2, Smad3, and p38.

Wei et al [56] first reported that Geniposide reduced the inflammatory response and oxidative stress and inhibited NLRP3 inflammasome activation in IPF mice, and used metabolomics analysis to prove that the mechanism may be linked to the regulation of host metabolism. Our results further confirmed the protective effect of Geniposide against pulmonary fibrosis, and we also found that Geniposide both inhibited the inflammatory response in the early stage and the fibrotic process in the late stage of pulmonary fibrosis. Mechanistic results showed that TGF- β 1, Smad2, Smad3, and p38 were the targets of Geniposide, and the protective effect of Geniposide against PF was related to inhibition of the key signaling pathways including TGF-β/Smad and p38MAPK signaling pathways in the development of PF.

## 5. Conclusions

In conclusion, our research suggests that Geniposide may have the potential to decrease the early inflammatory response and subsequent fibrosis in the lung tissues of mice with BLM-induced pulmonary fibrosis. This effect is possibly achieved by Geniposide inhibiting the TGF-β/Smad and p38MAPK signaling pathways, thus alleviating inflammatory and fibrotic processes. Compared to Gentiopicroside, another iridoid glycoside with similar properties, Geniposide has a more stable chemical structure and is found in abundance in dried Gardenia

fruits. Our study implies that Geniposide could be a promising candidate for the treatment of pulmonary fibrosis, suggesting the need for further research and development.

## Supporting information

**S1 Raw images. Western blot original image.**
(PDF)

**S1 Fig. Experimental program.**
(TIF)

**S2 Fig. HPLC curves of Geniposide.**
(TIF)

**S1 Data.**
(XLSX)

**S1 File. Histopathology scoring method.**
(PDF)

## Author Contributions

**Conceptualization:** Xuan Zhang.

**Data curation:** Jian-Bin Yin, Ying-Xia Wang, Su-Su Fan.

**Formal analysis:** Jian-Bin Yin, Ying-Xia Wang, Su-Su Fan.

**Funding acquisition:** Xuan Zhang.

**Methodology:** Jian-Bin Yin, Ying-Xia Wang, Su-Su Fan, Wen-Bin Shang, Yu-Shan Zhu, Xue-Rong Peng.

**Software:** Su-Su Fan.

**Supervision:** Cheng Zou, Xuan Zhang.

**Writing – original draft:** Jian-Bin Yin, Ying-Xia Wang, Su-Su Fan.

**Writing – review & editing:** Cheng Zou, Xuan Zhang.

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
