## [Decision Letter · Decision Letter 0]

28 Feb 2024

PONE-D-24-05055Geniposide ameliorates bleomycin-induced pulmonary fibrosis in mice by inhibiting TGF-β/Smad and p38MAPK signaling pathwaysPLOS ONE

Dear Dr. Zhang,

Thank you for submitting your manuscript to PLOS ONE. After careful consideration, we feel that it has merit but does not fully meet PLOS ONE’s publication criteria as it currently stands. Therefore, we invite you to submit a revised version of the manuscript that addresses the points raised during the review process.

We look forward to receiving your revised manuscript.

Kind regards,

Jian Hao

Academic Editor

PLOS ONE

Journal Requirements:

2.To comply with PLOS ONE submissions requirements, in your Methods section, please provide additional information regarding the experiments involving animals and ensure you have included details on (1) methods of sacrifice, (2) methods of anesthesia and/or analgesia, and (3) efforts to alleviate suffering

" This research was funded by the National Natural Science Foundation of China (No. 82260727),Yunnan Provincial Science and Technology Department (No.202101AY070001-010), and the Innovation Team Construction Project of Kunming Medical University（CXTD202203）."       

5. We note that your Data Availability Statement is currently as follows: "All relevant data are within the manuscript."

Additional Editor Comments :

Thank you for submitting your manuscript titled "Geniposide ameliorates bleomycin-induced pulmonary fibrosis in mice by inhibiting TGF-β/Smad and p38MAPK signaling pathways" . We have now completed the review process, and after careful consideration external reviewers, we have decided that your manuscript can only be considered for publication following major revisions.

The reviewers have acknowledged the importance and potential impact of your work. However, there are several areas that require significant improvement to meet the journal's standards for publication. Please find detailed comments and suggestions attached. If you have any concerns, please feel free let me know.

Reviewers' comments:

Reviewer's Responses to Questions

**Comments to the Author**

1. Is the manuscript technically sound, and do the data support the conclusions?

Reviewer #1: Yes

Reviewer #2: Yes

2. Has the statistical analysis been performed appropriately and rigorously? 

Reviewer #1: Yes

Reviewer #2: Yes

3. Have the authors made all data underlying the findings in their manuscript fully available?

Reviewer #1: Yes

Reviewer #2: Yes

4. Is the manuscript presented in an intelligible fashion and written in standard English?

Reviewer #1: Yes

Reviewer #2: Yes

5. Review Comments to the Author

Reviewer #1: The article investigates the effects of Geniposide, a natural iridoid glycoside derived from the fruits of Gardenia jasminoides Ellis, on pulmonary fibrosis induced by bleomycin in mice. The study examines the impact of Geniposide on chronic inflammation and fibrosis, as well as its mechanism of action through the inhibition of the TGF-β/Smad and p38MAPK signaling pathways.

The study addresses an important medical condition, pulmonary fibrosis, for which treatment options are limited.

However,

1) The specific dosages and treatment duration of Geniposide are not provided in abstract.

2) Insufficient justification for Geniposide selection: The introduction mentions that Geniposide has various pharmacological activities, including anti-inflammatory effects, but it does not provide a strong justification for selecting Geniposide as a potential therapeutic agent for PF. A more detailed explanation of the rationale behind this choice would strengthen the introduction.

3) The description of the histopathology scoring method is incomplete. It refers to the scoring method of Szapield and Fulmer et al., but the specific criteria used for grading alveolitis and fibrosis are not provided. Including the criteria and providing examples would enhance the reproducibility of the results.

4) In the 2.10 Molecular docking section, add more details. For example, 90×90× 90Å (x, y, and z) grid box protease binding pocket with XXX nm spacing for each dimension and a grid center at dimensions of XXX, YYY and ZZZ.

5) In result section, add degree of freedom for data as well as exact p value. For example: [F(4,36)=100, p=0.003]

6) First paragraph of discussion just mention the most important findings.

7) Additional limitations of the study design could be acknowledged, such as the use of only male mice. Discussion of these limitations would add balance.

8) Compare more about 14th day and 28th days.

9) The writing could be tightened in places to increase clarity and readability. For example, some sentences are quite long. Breaking them up may help.

Good Luck.

Reviewer #2: In this paper, Zhang et al. examine the effects of geniposide on beomycin induced fibrosis. They found geniposide can reduce lung fibrosis and inflammation in a dose dependent manner. They also found geniposide can down-regulate the expression of TGF-β1, Smad2/3, p38, and CTGF. Although their data looks convincing, the novelty and the molecular/cellular mechanisms underlying this effect need to be improved.

Major issues:

1. They claimed there are no reports of whether geniposide has a protective effect against pulmonary fibrosis (page 5, line 97), which is not true. Yi Wei et al reported similar effect and published their results in Journal of Functional Foods, Volume 104, May 2023, 105503. They found that geniposide improves bleomycin-induced pulmonary fibrosis by inhibiting NLRP3 inflammasome activation and modulating metabolism. Zhang will need to discuss and compare the difference/novelty between their study with this study.

2. They conclude from H&E staining that geniposide (Gen) significantly reduces the number of inflammatory cells, including macrophage (M), granulocyte (G) and lymphocyte. The quantification for this data is missing. They should use arrows to point out examples of macrophage (M), granulocyte (G) and lymphocyte in their images. Immunostaining of marker genes for these immune cells will strengthen their findings. Also, resident macrophages change their morphology upon activation. Morphological characterization of the morphology macrophages will help as well.

3. They showed that geniposide reduces inflammation and bleomycin-induced idiopathic pulmonary fibrosis. Does geniposide directly act on immune cells to inhibit their reactivity, which then contributes to reduction of fibrosis. In vitro culture of macrophages treated with geniposide can support this hypothesis.

Miner issues:

1. HPLC curves to show the purity of geniposide should be provided. (page 6)

2. Quantification results should be in the same figure with their representative. This applies to figure 1 to 3.

3. Representative image are usually put in front of quantification results. This applies to fig.6 to 7.

4. Molecular docking predicts geniposide binds to TGF-β1, Smad2, Smad3, and p38. Are their any proteins which geniposide does not bind? Showing a few examples of proteins which geniposdie does not bind will highlight the specificity of this drug.

5. Error bars should be provided in staining images. (Figure 1 and 2)

6. Are there any bad effects of geniposide treatment, such as cell death, tissue damage, or organ failure?

6. PLOS authors have the option to publish the peer review history of their article (what does this mean?). If published, this will include your full peer review and any attached files.

Reviewer #1: **Yes: **Ahmad Reza Dehpour

Reviewer #2: No

---

## [Author Response · Author response to Decision Letter 0]

4 Aug 2024

Reviewer #1:

1) The specific dosages and treatment duration of Geniposide are not provided in abstract.

Response(R):The modifications have been completed. Please refer to the red font section in the “Abstract”.

2) Insufficient justification for Geniposide selection: The introduction mentions that Geniposide has various pharmacological activities, including anti-inflammatory effects, but it does not provide a strong justification for selecting Geniposide as a potential therapeutic agent for PF. A more detailed explanation of the rationale behind this choice would strengthen the introduction.

R:The modifications have been completed. Please refer to the red font section in the “Introduction:“Our previous research found that Geniopicoside, which also has anti-PF effects[25], has a similar structure to Gen and belongs to the class of iridoid glycosides. However, Gen has a more stable structure and is easier to obtain in nature than Geniopicoside. Based on these, we hypothesized that Geniposide may have a potential therapeutic effect on PF.”

3) The description of the histopathology scoring method is incomplete. It refers to the scoring method of Szapield and Fulmer et al., but the specific criteria used for grading alveolitis and fibrosis are not provided. Including the criteria and providing examples would enhance the reproducibility of the results.

R: The histopathological scoring method has been added to the "Supporting information" section. Please refer to the "S4 Histopathology scoring method"

4) In the 2.10 Molecular docking section, add more details. For example, 90×90× 90Å (x, y, and z) grid box protease binding pocket with XXX nm spacing for each dimension and a grid center at dimensions of XXX, YYY and ZZZ.

R: The details of docking parameters have been added, Please refer to 2.10 Molecular docking section. 

5) In result section, add degree of freedom for data as well as exact p value. For example: [F(4,36)=100, p=0.003]

R: The modifications have been completed.

6) First paragraph of discussion just mention the most important findings.

 R:The modifications have been completed. Please refer to the first paragraph of the "Discussion" section

7) Additional limitations of the study design could be acknowledged, such as the use of only male mice. Discussion of these limitations would add balance.

R: The modifications have been completed. Please refer to the second paragraph of the "Discussion" section

8) Compare more about 14th day and 28th days.

R: It cannot be directly compared because 14 and 28 days are animals euthanized in different time batches, and they are in different stages of pulmonary fibrosis disease. The pulmonary fibrosis model mice were euthanized from the first day of tracheal instillation of bleomycin until the 28th day, which was a continuous and progressive process. In the experimental design of this study, lung tissue was selected to be taken on the 14th and 28th days respectively, because on the 14th day, inflammation was mainly present in the lung tissue and it was a stage of transition toward fibrosis; On the 28th day, significant fibrosis had formed in the lung tissue, and the inflammatory response had gradually subsided, manifested as chronic inflammation.

Reference:+

[1]Liu T, De Los Santos FG, Phan SH. The Bleomycin Model of Pulmonary Fibrosis. Methods Mol Biol. 2017;1627:27-42. doi: 10.1007/978-1-4939-7113-8_2. PMID: 28836192.

9) The writing could be tightened in places to increase clarity and readability. For example, some sentences are quite long. Breaking them up may help.

R: Some long sentences have been break into short sentences.

Reviewer #2:

Major issues:

1. They claimed there are no reports of whether geniposide has a protective effect against pulmonary fibrosis (page 5, line 97), which is not true. Yi Wei et al reported similar effect and published their results in Journal of Functional Foods, Volume 104, May 2023, 105503. They found that geniposide improves bleomycin-induced pulmonary fibrosis by inhibiting NLRP3 inflammasome activation and modulating metabolism. Zhang will need to discuss and compare the difference/novelty between their study with this study.

R: Thanks for kindly reminding us of this reference, we didn’t find this reference before. We have deleted the claim of no reports of whether geniposide has a protective effect against pulmonary fibrosis. And we discussed and compared the difference/novelty between our study and this reference in the Discussion Section.

2. ①They conclude from H&E staining that geniposide (Gen) significantly reduces the number of inflammatory cells, including macrophage (M), granulocyte (G) and lymphocyte. The quantification for this data is missing. ②They should use arrows to point out examples of macrophage (M), granulocyte (G) and lymphocyte in their images. ③Immunostaining of marker genes for these immune cells will strengthen their findings. ④Also, resident macrophages change their morphology upon activation. Morphological characterization of the morphology macrophages will help as well.

R①:In the original paper version, Fig 4 quantified the number of inflammatory cells in BALF, where Fig 4-A and B represent the total number of inflammatory cells in BALF, and Fig 4-C and D represent the classification and counting results of three types of inflammatory cells (M/G/L).

In the latest revised manuscript, we have merged Fig 3 and Fig 4 together. Please refer to "Fig 3" in the revised manuscript. 

R②: The modification has been completed, please refer to "Fig3"

R③:Sorry, for the convenience of observation and preservation under the microscope after staining the cell smear, we have undergone resin sealing treatment and cannot conduct further experiments; The cycle of preparing animal models again is relatively long.

R④:The modifications have been completed. We have described the morphology of macrophages after activation, please refer to the fifth paragraph of the "Discussion" section

3. They showed that geniposide reduces inflammation and bleomycin-induced idiopathic pulmonary fibrosis. Does geniposide directly act on immune cells to inhibit their reactivity, which then contributes to reduction of fibrosis? In vitro culture of macrophages treated with geniposide can support this hypothesis.

R: We have supplemented the in vitro study results. We observed the effect of geniposide on LPS-induced Raw264.7 macrophages and found that geniposide significantly changed the morphology of LPS induced Raw264.7 macrophages and inhibited the levels of IL-1β and TNF-α in cell supernatants，suggesting that geniposide could directly inhibit the activation of macrophage induced by LPS. please refer to “2.11 In vitro experiment” and “3.9 Effect of Geniposide on the activation of Raw 264.7 cells induced by LPS”

Minor issues:

1. HPLC curves to show the purity of geniposide should be provided. 

R: The HPLC curves were provided in the supporting information(S5).

2. Quantification results should be in the same figure with their representative. This applies to figure 1 to 3. 

R:We have merged Fig3 and Fig4 from the old version together, please refer to "Fig3" in the revised manuscript.

3. Representative image are usually put in front of quantification results. This applies to fig.6 to 7.

R: The modification has been completed, please refer to "Fig6, 7"

4. Molecular docking predicts geniposide binds to TGF-beta;1, Smad2, Smad3, and p38. Are their any proteins which geniposide does not bind? Showing a few examples of proteins which geniposdie does not bind will highlight the specificity of this drug.

R: In molecular docking, the binding energy score is used to evaluate the binding specificity of drug to the target proteins. The binding energy score criterion has been added in the 2.10 Molecular docking section. And in the 3.8 Molecular docking section, we added “The binding energy score was used to evaluate the binding specificity, and a score less than -5 was usually a criterion for specific binding. The binding energy scores of TGF-β1, Smad2, Smad3, and p38 were -8.6, -6.2, -6.8 and -6.9, respectively. ”

5. Error bars should be provided in staining images. (Figure 1 and 2)

R: We guess the reviewer meant the scale bars， and have added scale bars in in staining images （Fig2 and Fig3）

6. Are there any bad effects of geniposide treatment, such as cell death, tissue damage, or organ failure?

R: An article reported an acute toxicity test of geniposide [1,2], which showed that liver toxicity was only induced when the dose of geniposide reached 574mg/kg in rats. In another study [3], the authors induced a Kunming strain mouse liver injury model with different doses of geniposide; The results showed that when the dose reached 420mg/kg, there was a difference in serum ALT and AST levels between the mice and the normal group. From the above evidence, it can be seen that the dose range used in this study is safe, and no adverse reactions were observed during our research process.

[1]Ding, Y.; Zhang, T.; Tao, J.S.; Zhang, L.Y.; Shi, J.R.; Ji, G. Potential hepatotoxicity of geniposide, the major iridoid glycoside in dried ripe fruits of Gardenia jasminoides (Zhi-zi). Nat. Prod. Res. 2013, 27, 929–933.

[2]Shan, M.; Yu, S.; Yan, H.; Guo, S.; Xiao, W.; Wang, Z.; Zhang, L.; Ding, A.; Wu, Q.; Li, S.F.Y. A Review on the Phytochemistry, Pharmacology, Pharmacokinetics and Toxicology of Geniposide, a Natural Product. Molecules 2017, 22, 1689. https://doi.org/10.3390/molecules22101689

[3] Liu, Q.; Lu, B.P.; Jia, R. Preliminary study on the establishment of mouse model of liver injury by perfusing stomach with geniposide. China J. Chin. Med. 2013, 28, 994–996.

---

## [Decision Letter · Decision Letter 1]

12 Aug 2024

PONE-D-24-05055R1Geniposide ameliorates bleomycin-induced pulmonary fibrosis in mice by inhibiting TGF-β/Smad and p38MAPK signaling pathwaysPLOS ONE

Dear Dr. Zhang,

Thank you for submitting your manuscript to PLOS ONE. After careful consideration, we feel that it has merit but does not fully meet PLOS ONE’s publication criteria as it currently stands. Therefore, we invite you to submit a revised version of the manuscript that addresses the points raised during the review process.

We look forward to receiving your revised manuscript.

Kind regards,

Jian Hao

Academic Editor

PLOS ONE

Journal Requirements:

Reviewers' comments:

Reviewer's Responses to Questions

**Comments to the Author**

1. If the authors have adequately addressed your comments raised in a previous round of review and you feel that this manuscript is now acceptable for publication, you may indicate that here to bypass the “Comments to the Author” section, enter your conflict of interest statement in the “Confidential to Editor” section, and submit your "Accept" recommendation.

Reviewer #1: All comments have been addressed

Reviewer #2: All comments have been addressed

2. Is the manuscript technically sound, and do the data support the conclusions?

Reviewer #1: Yes

Reviewer #2: Yes

3. Has the statistical analysis been performed appropriately and rigorously? 

Reviewer #1: Yes

Reviewer #2: Yes

4. Have the authors made all data underlying the findings in their manuscript fully available?

Reviewer #1: Yes

Reviewer #2: Yes

5. Is the manuscript presented in an intelligible fashion and written in standard English?

Reviewer #1: Yes

Reviewer #2: Yes

6. Review Comments to the Author

Reviewer #1: (No Response)

Reviewer #2: The authors have addressed most of my concerns, resulting in a significant improvement in the quality of the paper after revision. However, there are still a few minor issues that need attention:

Minor Issues:

Figure 2: Scale bars are not very visible and appear too thin. Consider using thicker lines and red instead of black for better contrast. Additionally, please clarify what the red feature is in the H&E staining of the PFD group in the right panel as this feature seems unique to the PFD group.

Figure 3: It remains unclear how different types of immune cells are recognized in the H&E staining. Please include the criteria used for this seperation in the Methods section. Additionally, the total number of cells in the model group of BALF doubles at 28 days compared to 14 days, but the number of immune cells (M, L, G) does not reflect this change. Please discuss which cell types contribute to this significant increase in cell numbers.

Figure 6: Please explain why there is a significant increase in TGF-beta levels in the Gen25 group compared to the model group, while Gen50 shows a reduction in TGF-beta levels.

7. PLOS authors have the option to publish the peer review history of their article (what does this mean?). If published, this will include your full peer review and any attached files.

Reviewer #1: **Yes: **Ahmad Reza Dehpour

Reviewer #2: No

---

## [Author Response · Author response to Decision Letter 1]

16 Aug 2024

Reviewer # 2 question：

1.Figure 2: ①Scale bars are not very visible and appear too thin. Consider using thicker lines and red instead of black for better contrast. ②Additionally, please clarify what the red feature is in the H&E staining of the PFD group in the right panel as this feature seems unique to the PFD group. 

R：

①According to the suggestion, we use a red bold scale for identification

②We speculate that you are referring to the staining images of the PFD group in the Pathological H&E staining section of Fig 2 on day 28. The red features in this image represent pulmonary vessels; After H&E staining, the vascular contents (platelets, etc.) were stained red.It should be emphasized that this red feature is not unique to the PFD group, but when selecting the field of view under the microscope, other experimental groups happened to avoid the pulmonary vessels.

2.Figure 3: ①It remains unclear how different types of immune cells are recognized in the H&E staining. Please include the criteria used for this seperation in the Methods section. ②Additionally, the total number of cells in the model group of BALF doubles at 28 days compared to 14 days, but the number of immune cells (M, L, G) does not reflect this change. Please discuss which cell types contribute to this significant increase in cell numbers.

R：

①We mentioned in the footnote at Fig 3 of the manuscript: 'Based on the morphological characteristics of macrophage (M), granulocyte (G) and lymphocyte (L) nuclei, the number of three types of inflammatory cells (M/G/L) in each photograph was counted and the average was calculated (Set to bM/G/L)'. We have provided a detailed introduction to the criteria for identifying different immune cells in the "Methods" section of the manuscript.

②Thank you for reading our article thoroughly and asking such insightful questions. Our explanation is that in Fig 3, B-C represents the total number of cells actually counted in BALF; In the quantitative analysis of three types of cells (M/G/L) (D-E), in order to reduce bias, we use the proportion of three types of cells (M/G/L) to the total number of cells to represent the results. Since this value is calculated by the ratio of the number of immune cells (a) [M/G/L] to the total number of cells (b) during the same period (14/28 days) (i.e. a/b), the differences in the number of immune cells at different periods cannot be reflected in the histogram D \\ E.

3.Figure 6: Please explain why there is a significant increase in TGF-beta levels in the Gen25 group compared to the model group, while Gen50 shows a reduction in TGF-beta levels.

R：In Fig 6, B represents the expression of TGF beta in lung tissue on day 14. From the graph, it can be visually observed that the Gen 25 group appears to be significantly upregulated compared to the model group, but after statistical calculations, we found that the difference between the two groups was not significant (P>0.05); Therefore, it cannot be considered that the Gen 25 group is significantly upregulated. As for the downregulation of Gen 50 group expression, we believe it is due to the action of the drug.

---

## [Decision Letter · Decision Letter 2]

20 Aug 2024

栀子苷通过抑制 TGF-β/Smad 和 p38MAPK 信号通路改善博来霉素诱导的小鼠肺纤维化

PONE-D-24-05055R2

Dear Dr. Zhang,

We’re pleased to inform you that your manuscript has been judged scientifically suitable for publication and will be formally accepted for publication once it meets all outstanding technical requirements.

Kind regards,

Jian Hao

Academic Editor

PLOS ONE

Additional Editor Comments (optional):

Reviewers' comments:

Reviewer's Responses to Questions

**Comments to the Author**

1. If the authors have adequately addressed your comments raised in a previous round of review and you feel that this manuscript is now acceptable for publication, you may indicate that here to bypass the “Comments to the Author” section, enter your conflict of interest statement in the “Confidential to Editor” section, and submit your "Accept" recommendation.

Reviewer #2: All comments have been addressed

2. Is the manuscript technically sound, and do the data support the conclusions?

Reviewer #2: Yes

3. Has the statistical analysis been performed appropriately and rigorously? 

Reviewer #2: Yes

4. Have the authors made all data underlying the findings in their manuscript fully available?

Reviewer #2: Yes

5. Is the manuscript presented in an intelligible fashion and written in standard English?

Reviewer #2: Yes

6. Review Comments to the Author

Reviewer #2: They addressed all my concerns. I have no more comments to them. Congratulations to all the authors.

7. PLOS authors have the option to publish the peer review history of their article (what does this mean?). If published, this will include your full peer review and any attached files.

Reviewer #2: No

---

## [Editor Report · Acceptance letter]

27 Aug 2024

PONE-D-24-05055R2 

PLOS ONE

Dear Dr. Zhang, 

I'm pleased to inform you that your manuscript has been deemed suitable for publication in PLOS ONE. Congratulations! Your manuscript is now being handed over to our production team.

Kind regards, 

on behalf of

Dr. Jian Hao 

Academic Editor

PLOS ONE